# Cellular Response of Neutrophils to Bismuth Subnitrate and Micronized Keratin Products In Vitro

**DOI:** 10.3390/vetsci7030087

**Published:** 2020-07-06

**Authors:** Shirli Notcovich, Norman B. Williamson, Jimena Yapura, Ynte Schukken, Cord Heuer

**Affiliations:** 1School of Veterinary Science, Massey University, Palmerston North 4410, New Zealand; N.Williamson@massey.ac.nz (N.B.W.); M.J.Yapura@massey.ac.nz (J.Y.); C.Heuer@massey.ac.nz (C.H.); 2Department of Animal Sciences, Wageningen University, 6700 Wageningen AH, The Netherlands; ynte.schukken@wur.nl; 3GD Animal Health, 7400 Deventer AA, The Netherlands

**Keywords:** teat sealants, bismuth subnitrate, keratin, neutrophils, teat canal, immune response

## Abstract

The aim of this study was to assess the effect of bismuth subnitrate and micronized keratin on bovine neutrophils in vitro. We hypothesized that recruitment and activation of neutrophils into the teat canal and sinus are the mechanisms of action of bismuth subnitrate and keratin-based teat sealant formulations. To test this, a chemotaxis assay (Experiment 1) and a myeloperoxidase (MPO) assay (Experiment 2) were conducted in vitro. Blood was sampled from 12 mid-lactation dairy cows of variable ages. Neutrophils were extracted and diluted to obtain cell suspensions of approximately 10^6^ cells/mL. In Experiment 1, test substances were placed in a 96-well plate, separated from the cell suspension by a 3 µm pore membrane and incubated for 3 h to allow neutrophils to migrate through the membrane. In Experiment 2, neutrophils were exposed to the test products and the amount of MPO released was measured by optical density. Results showed that neutrophils were not activated by bismuth or keratin products (*p* < 0.05) in all of the tests performed. These results suggest that the mechanisms of action of bismuth subnitrate and keratin-based teat sealants do not rely on neutrophil recruitment and activation in the teat canal and sinus after treatment.

## 1. Introduction

Methods to treat and prevent mastitis in the dry period have been available since the middle of last century [1,2]. The current and most commonly used methods are teat sealants and dry cow antibiotic therapy. The use of these formulations reduced the incidences of intramammary infections in the dry period and early lactation in treated compared to non-treated quarters [3]. Teat sealants containing bismuth subnitrate were developed in the 1970s and are effective at preventing new dry period infections in challenge and natural exposure studies [4,5]. The proposed mechanism of action of bismuth-based products is the creation of a physical barrier that prevents passage of mastitis pathogens into the teats, but this has not been shown experimentally despite their proven efficacy.

The teat canal is the duct that connects the lumen of the udder with the external environment. It is lined by modified teat skin. Macrophages, Langerhans, plasma, and dendritic cells have been described as being present under the basal membrane in the teat canal epithelial tissue. They are dedicated cells that take up, process, and present antigens to T-cells in their major histocompatibility complex (MHC) receptors during infections [6,7]. Neutrophils are polymorphonuclear cells (PMN), which are the first active line of defense against pathogens entering the mammary gland. The surveillance roles conducted by PMN consist mainly of their ability to migrate to the site of infection (chemotaxis), a respiratory burst and myeloperoxidase (MPO) release. During phagocytosis, neutrophils increase their oxygen consumption through the activity of nicotinamide adenine dinucleotide phosphate oxidase (NADPH-oxidase), and via successive electron reductions generate superoxide anion O_2_ and hydrogen peroxide (H_2_O_2_) [8,9]. This process is known as the respiratory burst. These oxygen metabolites activate additional reactive oxygen species (ROS) that are strongly anti-microbial, for example MPO. Myeloperoxidase is an enzyme present in azurophilic granules in neutrophils and performs a vital role in destroying phagocytosed microorganisms. In mastitis, neutrophils migrate from blood at the Fürstenberg’s rosette and teat sinus into the teat lumen, following chemotactic signals released from these antigen presenter cells, bacterial invaders or damaged tissue [10,11]. The presence of high somatic cell counts (SCC) in milk has a protective effect against mastitis induced by major pathogens [12,13,14]. The presence of neutrophils in milk could be considered valuable in preventing infection by major mastitis pathogens, especially during the early dry period and after treatment with internal teat sealants. However, most of the reviewed literature shows that an increase in somatic cell counts (PMN, macrophages, and desquamating epithelial tissue) are signs of intramammary infection [15,16]. A chemotactic effect of bismuth subnitrate based teat sealant has not been experimentally demonstrated. Based on this, another proposed mechanism of action of bismuth subnitrate based teat sealants is the generation of a local cellular immune response after treatment that protects and, hypothetically, cures subclinical mastitis [17]. Similarly, a novel keratin-based teat sealant product under development showed, in preliminary studies, an increase in SCC after treatment suggestive of a mechanism of action related to the recruitment of neutrophils into the teat canal and sinus [18].

The hypothesis of this study is that bismuth subnitrate and keratin-based teat sealants induce chemotaxis and activation of neutrophils in the teat canal and sinus and hence protect the mammary gland against mastitis infections during the early dry period. In order to test this hypothesis, our objective was to evaluate the ability of bismuth subnitrate and micronized keratin to induce a cellular response (migration and activation) in vitro.

## 2. Materials and Methods

### 2.1. Blood Collection and Neutrophil Preparation from Peripheral Blood

The study protocol was approved by the Massey University Animal Ethics Committee, protocol number 18/02, on the 26 February 2018 Fifty-milliliter blood samples were collected into EDTA tubes (BD-vacutainer K2E EDTA, BD-Plymouth, PL6 7BP, UK) by coccygeal venipuncture of 12 dairy cows at mid-lactation. This study compared the in vitro activation of neutrophils in response to bismuth subnitrate and micronized keratin against a negative control using media only.

Neutrophils were isolated from blood and the number of neutrophils per sample was determined using a TC20 automated cell counter, (Bio-Rad Laboratories, Hercules, CA 94547, USA). Neutrophil suspensions were used to complete chemotaxis assays and myeloperoxidase tests in vitro. Within 1 h of blood collection, neutrophils were separated from whole blood samples. Briefly, approximately 50 mL of blood was transferred into two 50 mL conical tubes and centrifuged at 1500× *g* for 30 min at room temperature. After centrifugation, the plasma, buffy coat layer and top layer (approximately 1/3) of packed red blood cells were removed by aspirating with a Pasteur pipette. Approximately seven milliliters of blood were left in each tube and 38 mL of MQ water was added to lyse the red blood cells. The tube was rotated for 5 s. Then, 5 mL of 10× *g* concentrated PBS pH 7.4 was added and the tube immediately rotated again to restore osmotic balance for the cells. The tubes were centrifuged at 330× *g* for 10 min at room temperature (20–22 °C) and the supernatant discarded, leaving a neutrophils-rich red pellet at the bottom of each tube. The pellet was washed with 10 mL of 1 × PBS pH 7.4 and vortexed to mix. The tubes were centrifuged for 5 min at 670× *g* at room temperature 20–22 °C, the supernatant was discarded, and the cells re-suspended in 2 mL of the media required in the procedures for each performed assay. Cell suspensions prepared using this procedure had ≥90% pure populations of living neutrophils confirmed by cell counting with trypan blue.

### 2.2. Chemotaxis Assay (Experiment 1)

A chemotaxis assay was performed using a CytoSelect 96-well Cell Migration Assay (3 µm, Fluorometric Format, Cell Biolabs. Inc., San Diego, CA 92126, USA). A cell suspension containing approximately 5 × 10^6^ cells/mL was prepared in serum free-media (RPMI 1640, R8758-500ML, Sigma Aldrich, Inc. Christchurch, New Zealand) containing 0.5% bovine serum albumin (BSA). Fetal bovine serum 10% was used as a positive control chemoattractant. Fetal bovine serum 10% was used as a chemotactic agent for the positive control (PC) and 0.5% BSA-RPMI was used as the negative control (NC). Keratin and bismuth subnitrate suspensions were prepared as follows to create high and low concentrations (3% and 1.5%, respectively). These concentrations were chosen, as 3% was the maximum concentration of bismuth subnitrate that could be manipulated with minimal precipitation of the suspension. For the keratin and bismuth suspensions in high concentrations (KH and BH treatments), 1.2 g of bismuth subnitrate or sterile micronized keratin were added to 40 mL of RPMI. Keratin and bismuth at low concentrations (KL and BL treatments) were prepared with 0.6 g of bismuth subnitrate or micronized keratin in 40 mL of RPMI. In a cell culture chamber, 150 µL of KH, BH, KL, BL, the positive control and RPMI with 0.5% BSA used as the negative control (NC) were loaded in the feeder tray (bottom plate) in triplicates. A membrane chamber was placed on top of the feeder tray. One hundred microlitres of the cell suspension were added to each well and the plate was incubated in a cell culture incubator at 37 °C for 3 h. The cell suspension from inside the membrane chamber was carefully discarded by inverting the plate and the insert chamber was transferred to a clean 96-well plate containing 150 µL of pre-warmed Cell Detachment Solution. This plate was incubated for 30 min at 37 °C. Cells were completely dislodged from the underside of the membrane by gently tilting the insert several times in the Detachment Solution, and the insert was removed and discarded. Seventy-five microlitres of media from the feeder tray were combined with 75 µL of the Detachment Solution in a clean 96-well plate. Fifty microlitres of 4X Lysis Buffer/CyQuant GR dye solution was added to each well (already containing 150 µL of Cell Detachment Solution). This step combines cells that migrated through the membrane and into the medium and migratory cells that were detached from the bottom side of the membrane by the Cell Detachment Solution. The plate was incubated for 20 min at room temperature. One hundred and fifty µL of the mixture was transferred to a 96-well plate suitable for fluorescence measurement. Fluorescence was read using a fluorescence plate reader (Flexstation 3 by Molecular Devices, Biostrategy, Auckland, New Zealand) at 480 nm/520 nm (SoftMaxPro version 5.4.1, Molecular Devices, Biostrategy, Auckland, New Zealand). Fluorescence levels in this study were obtained as a result of the lysis of migrated cells in the wells at the bottom of the analyzed plate.

### 2.3. Myeloperoxidase Assay (Experiment 2)

Myeloperoxidase release was measured in a 96-well plate from Phorbol 12-myristate 13-acetate (PMA, positive control), keratin, or bismuth-stimulated neutrophils and from lysed neutrophils to account for the total amount of MPO contained within the neutrophils. Isolation of neutrophils was performed as described above obtaining 2.5 × 10^7^ cells/mL of HBSS (14025092, Life Technologies NZ Ltd. Auckland, New Zealand). A 96-well plate (ELISA Microplate 96F, GR655061, Grenier Bio-one, International GmbH, Kremsmünster, Austria) was seeded with 50 µL/well of lysis reagent, 50 µL/well of stimulation reagent or 50 µL/well of bismuth or keratin in high concentration (3% BH, KH). Negative control (NC) wells contained 50 µL/well of HBSS. The lysis reagent contained 0.02% hexadecyltrimethylammonium bromide (Sigma) in MQ water. Stimulation reagent was prepared by mixing 1 part of CaI stock solution (50 μg/mL calcium ionophore A23187 (Sigma) in HBSS), 1 part of cytochalasin B stock solution (50 μg/mL cytochalasin B (Sigma) in HBSS), 1 part of PMA stock solution (20 μg/mL PMA (Sigma) in HBSS, and 7 parts of HBSS. Plates were incubated at 30 °C for 60 min. 3,3′,5,5′-Tetramethylbenzidine (Becton Dickinson Ltd. Auckland, New Zealand) and hydrogen peroxide were mixed 1:1, and 100 μL of the mixture was added to each well. Colour was allowed to develop at room temperature for approximately 2 min and 50 μL of 2 N H_2_SO_4_ was added as a stop reagent. Plates were centrifuged at 600× *g* for 5 min at room temperature, and 150 μL/well was transferred into a new plate. Optical density at 450 nm was determined using a microtiter plate spectrophotometer (Versamax, Molecular Devices, Sunnyvale, CA, USA).

### 2.4. Statistical Analysis

The data from these studies were not normally distributed, so they were transformed by log2 transformation. Mixed linear models included treatment as a fixed effect and cow as a random effect in SAS version 9.4 (SAS Institute Inc., Cary, NC, USA). Statistical significance was set at *p* < 0.05. Statistical differences were based on Tukey-Kramer adjusted pairwise comparison per product tested against the negative control.

## 3. Results

### 3.1. Chemotaxis Assay (Experiment 1)

Bismuth subnitrate and keratin products did not show signs of increased chemotaxis (Figure 1). Bismuth subnitrate in high (BH) and low concentrations (BL) showed significantly lower levels of chemotaxis than the negative control.

### 3.2. Myeloperoxidase Assay (Experiment 2)

Bismuth and keratin induced a lower release of MPO than the negative control (*p* < 0.001, Figure 2). Negative control (NC) values represent approximately 50% of the values obtained by the total lysis of the neutrophils and their total MPO content (Figure 2, Lysis).

## 4. Discussion

In this study we investigated the response of bovine neutrophils to bismuth subnitrate and micronized keratin products in vitro. We hypothesized that these products activate neutrophils inducing chemotaxis and MPO release. However, the results show that neutrophils were not activated by the test compounds in vitro. Most of the research on the mechanism of action of bismuth focuses on the mechanical effect of bismuth subnitrate intramammary teat sealants in preventing bacterial infections by blocking the teat canal and teat sinus to bacterial colonization [3,19]. This appears to be the first published study to evaluate in vitro the cellular response as a potential mechanism of action of bismuth subnitrate and the novel micronized keratin compound.

The objective of Experiment 1 was to evaluate in vitro the chemotactic effect of bismuth subnitrate and micronized keratin on bovine neutrophils. Previous studies have proposed that the mechanism of action of bismuth subnitrate might be the activation of immune cells in the teat canal [17,20]. Bismuth subnitrate used in internal teat sealants, with or without concomitant antimicrobial treatment, prevented and cured mastitis by generating an increase in neutrophil concentration and cytokine expression in the mammary gland [17]. These results showed that bismuth subnitrate treated teats had an increased number of neutrophils in milk after treatment and a significant increase in the TNF-α:IL-8 ratio. However, care must be taken when analyzing these results as the study had its limitations. The increase in neutrophils occurred in all treated teats whether treated with dry cow therapy or bismuth. Hence, there was no evidence to conclude that bismuth subnitrate induced neutrophil migration. In addition, the study did not describe how teat sealants were applied to teats. Any substance infused into the mammary gland may induce an immune response mainly involving PMN. Constricting the teat sinus at the base of the teat when infusing internal teat sealants is important, as it avoids the products being infused into the mammary tissue and inducing a large cellular response. The application of a chitosan-based teat sealant product, when compared with a bismuth-based teat sealant and an untreated control, did not show differences in neutrophil concentration or in markers of inflammation in the milk of treated cows [20]. This supports a lack of chemotactic effect in vitro from bismuth subnitrate and chitosan as was shown in the current study for bismuth subnitrate and micronized keratin. Our results align with others who found that chemotaxis of macrophages was inhibited by tripotassium dicitrato bismuthate (a bismuth-based treatment for stomach ulcers), thus showing a putative anti-inflammatory effect of bismuth-based formulations [21].

We found that a small percentage of neutrophils migrated through the 3 µm membrane (NC was different from zero). This could have been caused by random migration through the membrane or simply by gravity, which has been reported previously [22]. The significantly lower values of chemotaxis shown by the bismuth-exposed neutrophils compared to the negative control could be an indication that bismuth subnitrate prevented or inhibited the migration of neutrophils through the membrane. This could be explained by bismuth subnitrate being a charged molecule. Chemotaxis starts by depolarization of the neutrophil’s membrane followed by a period of hyperpolarization [23]. Bismuth subnitrate charged molecules may interact with the neutrophil membranes to prevent depolarization or prolong the hyperpolarization period, thus impeding the passage of the neutrophils through the 3 µm pores. It is also possible that bismuth subnitrate particles blocked the pores, as in our study BH (bismuth in high concentration, 3%) showed lower levels of chemotaxis than BL (1.5%). In further research a different experimental design would be required to demonstrate such mechanisms. There was no difference between the results of keratin in high and low concentrations and the negative control in this study, suggesting that keratin did not induced chemotaxis in vitro. This is opposite to what has been reported in previous in vivo studies, where an increase in SCC was present after introducing a micronized keratin product into the teat canal [18].

Myeloperoxidase release from neutrophils exposed to bismuth and keratin was significantly lower than from the negative controls that were not exposed. One of the biggest limitations of this study was that the positive control failed to activate the neutrophils. Exposure of neutrophils to cytochalasin B, a fungal metabolite, blocks the polymerization of contractile microfilaments and, consequently, facilitates degranulation [24]. However, even though neutrophils were treated with cytochalasin B and PMA in this study, they did not show signs of MPO being released in the positive control wells. Although in this study PMA-stimulated cells were not significantly different from the NC, the MPO value of the cells that were lysed (thus releasing 100% of the MPO content in the cells) was considered a valid positive control. All the other values, therefore, can be presented as a proportion of this 100% MPO content since all showed significantly lower levels of MPO release [25]. Since bismuth and keratin presented significantly lower levels of MPO release than the positive and negative control we can confidently conclude that there was no increased activation of neutrophils. The lack of reaction to the PMA could be due to an early activation and exhaustion of MPO within the cells prior to the conduct of the study due to manipulation as this has been reported in previous studies [26,27].

There is evidence that bismuth compounds (specifically bismuth subsalicylate and bismuth subgallate) have anti-inflammatory effects. Bismuth subgallate inhibited nitric oxide production in macrophages by inhibiting the mRNA expression and stability of the enzyme nitric oxide synthases [28]. In addition, a reduction in mononuclear cells and neutrophilic inflammation was observed in the histological analysis of patients with microscopic colitis after treatment with bismuth subsalicylate [29]. This could explain the low values of MPO observed in this study for the bismuth exposed neutrophils. Historically, bismuth subnitrate was used by intramuscular injection for treatment of syphilis [30]. If bismuth subnitrate had induced chemotaxis and activation of neutrophils when injected intramuscularly, the site of injection would have been swollen and inflamed. However, no reports occur on bismuth subnitrate causing an inflammatory reaction at the injection site [31]. In agreement with the results obtained in this in vitro study where bismuth subnitrate did not induce chemotaxis or MPO release and activation after treatment, bismuth subnitrate is classified as an “inert product” when inserted into the teat sinus within a teat sealant product during the early dry period [32].

Studies on the effect of keratin products on the teat canal are lacking. Micronized keratin is currently used in the human cosmetic industry and there have been no reports of inflammatory reactions after treatment of skin with micronized or hydrolyzed keratin. Furthermore, in various studies conducted with micronized wool keratin applied to human skin, researchers found that the keratin formulations reinforced the skin barrier integrity and improved its water-holding capacity [33]. This supports our findings of low levels of activation and lack of chemotaxis for keratin exposed neutrophils [33,34].

The results obtained in this study should be complemented by in vivo experiments in which chemotaxis and activation of neutrophils within the teat canal are assessed. A reason for this is that differences between in vivo and in vitro study results exist, especially when dealing with inflammatory cell responses such as the mechanisms of activation of neutrophils that were addressed in this study [35].

These results provide an impetus for the continuing investigation of the mechanism of action of teat sealant treatments. For example, “what are the effects of bismuth subnitrate and keratin in vivo when inserted in the teat canal?” and “can bismuth or keratin prime neutrophils?” i.e., can bismuth or keratin take neutrophils to a more responsive state without activating them? These questions still remain unanswered. The results of this study strongly suggest that mechanisms of action of bismuth subnitrate, and micronized keratin are not the activation of a local immune response in the teat sinus and teat canal.

## 5. Conclusions

Results of the described in vitro experiments indicated that bismuth and keratin products did not induce chemotaxis nor activate neutrophils. Further in vivo research is recommended to address questions on the mechanisms of action of bismuth subnitrate and micronized keratin when utilized as internal teat sealants for prevention of intramammary infections.

## Figures and Tables

**Figure 1 vetsci-07-00087-f001:**
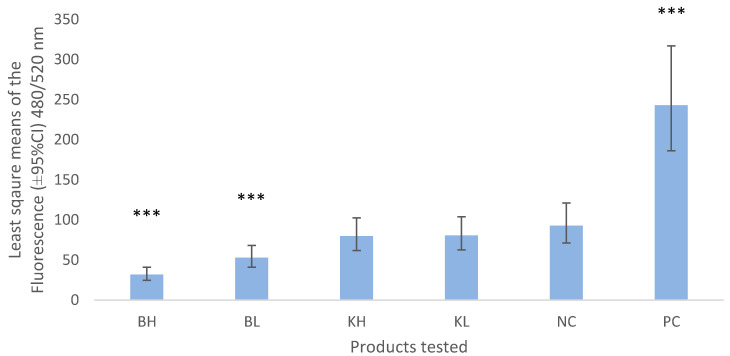
Least square means (±95% CI) of the fluorescence values obtained from chemotaxis assay measured at 480/520 nm wavelength. BH = bismuth high (3% *w/v*), BL = bismuth low (1.5% *w/v*), KH keratin high (3%), KL = keratin low (1.5%), NC = negative control (0.5% BSA-RPMI) PC = positive control (FBS). Asterisks show the significance levels of the comparison with the negative control. *** means *p*-value < 0.001.

**Figure 2 vetsci-07-00087-f002:**
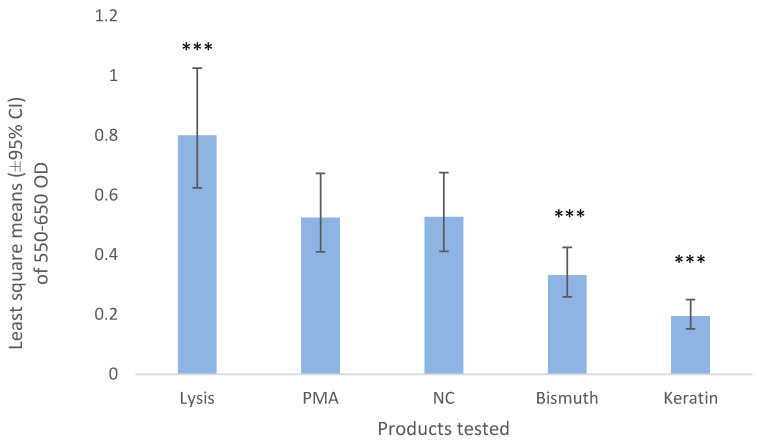
Least square means (±95% CI) of 550–650 optical density of the neutrophils exposed to bismuth 3% and keratin 3%. HBSS was used as a negative control (NC) and 1 part of 20 µg/mL of PMA, 1 part of 50 μg/mL calcium ionophore, 1 part of 50 μg/mL cytochalasin B with 7 parts of HBSS was used as the positive control (PMA). The lysis buffer contained 0.02% hexadecyltrimethylammonium bromide. *** means *p*-value < 0.001.

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
