# Peer review of "Cellular Response of Neutrophils to Bismuth Subnitrate and Micronized Keratin Products In Vitro"

_vetsci, 2020, doi:10.3390/vetsci7030087_

Round 1
Reviewer 1 Report
General comment
The paper deals with an interesting question and is carefully and clearly written. I have only some small comments concerning the form of the presentation. The significance of the study is limited, as also described by the authors. Nevertheless, a good first step to approach the clarification of this.
Specific comments:
Abstract: Experiment 1 + 2, capitalize consistently
Introduction: I would prefer the correct spelling of Fürstenberg with a nice “ü”
M&M, 2.1: sometimes there is an x written when giving the data for the centrifugation
M&M:2.2:
L 1: no space before Well
- xy: space before BSA
M&M, 2.3: consistent style for well plate needed
Discussion: new paragraph after Source No.32
Figures:
Diagram 1 is reminiscent of a box plot and is probably intended to show the median, maximum and minimum (or the standard deviation?). The only question I have is why the p-value is shifted to the right, the same with the 2nd diagram.
Diagram 2 is somewhat more confusing. The X-axis and the labeling of the individual tested products are missing, so that the bars cannot be assigned. The meaning of the small grey and black bars can only be guessed (different layout to diagram 1). In addition, the p-values cannot be clearly assigned.
References
- Smolenski, G.A. The bovine teat canal: Its role in pathogen recognition and defence of the mammary gland. (Thesis, Doctor of Philosophy (PhD)), The University of Waikato, Hamilton, New Zealand, 2018.
- Williamson, N.B. Methods for reducing the incidence of mastitis. 2012. https://patents.google.com/patent/WO2008020769A1/de (accessed on Day Month Year).
- Lanctôt, S.; Fustier, P.; Taherian, A.R.; Bisakowski, B.; Zhao, X.; Lacasse, P. Effect of intramammary infusion of chitosan hydrogels at drying-off on bovine mammary gland involution. Journal of Dairy Science 2017, volume, page range. doi: http://dx.doi.org/10.3168/jds.2016-12087
- Washburn, S.; Klesius, P.; Ganjam, V. Characterization of the chemiluminescence response of equine phagocytes. American Journal of Veterinary Research (USA) 1982, volume, page range.
- Fischer, T.W.; Wigger-Alberti, W.; Elsner, P. Assessment of ‘dry skin’: Current bioengineering methods nd test designs. Skin Pharmacology and Physiology 2001, 14, 183-195.
Reviewer 2 Report
This paper describes two in vitro studies that evaluate the potential effect of bismuth subnitrate and micronized keratin on neutrophil function. The content is likely to be of interest to readers in the field of primary physiology as well as some readers whose interest is of a more clinical emphasis.
This is a very well written paper. Overall, the standard of presentation, clarity of language and scientific argument is high.
Some specific comments:
Introduction:
The authors tend to use some jargon and technical terminology without adequate or appropriate explanation. For example, the concept of “respiratory burst” and “myeloperoxidase (MPO) release” is used early in the Introduction without being explained or referenced to explanatory literature. Whilst both concepts are explained later, the structure of the Introduction will be a little awkward to readers who are unfamiliar with these processes.
Some choices of words are not quite right. For example, do PMN “exert” a surveillance role? Are “Respiratory burst and MPO release” best described as “destruction mechanisms” and, if so, what are they destroying?
Results:
The authors state that bismuth subnitrate and keratin products “did not show signs of increased chemotaxis (Figure 1).” Indeed, the fluorescence of the BH and BL groups appear to be significantly lower than the negative control. Can this be interpreted as meaning that these agents are not only not increasing chemotaxis but may be inhibiting chemotaxis? The interpretation of these results could be expanded.
Both Figure 1 and Figure 2 have some formatting issues. The arrangement of the labeling of the vertical axes is clumsy. In Figure 2 the labeling of the x axis on my copy of the manuscript has totally disappeared, and there are some small bars which I assume are confidence intervals, but it isn’t at all clear.
Discussion:
Some terminology could be improved. For example, the term “action mechanism” is used repeatedly where I think “mechanism of action” is more grammatically correct and certainly is easier to read.
The text states “These results showed that bismuth subnitrate treated teats had increased neutrophils in milk after treatment…” This needs to be clarified. Does this mean that these teats had greater numbers of neutrophils? A higher concentration of neutrophils? Neutrophils of greater activity, or prolonged survival?
The authors recognise that the failure of the positive control to activate neutrophils is a substantial limitation of their study. As this is a fundamental aspect of the interpretation of their results, I would suggest that more effort is made to explain why the positive controls appeared to fail, and why this does not call into question the results of the study.
The meaning of the phrase “It is worthy to account for differences between in vivo and in vitro results,…” is not at all obvious to me.
The final sentences of the Discussion (beginning “These results provide an impetus…”) are poorly structured. It is not at all clear what point the authors are trying to make. As presented, this wording seems to be an indication that, having presented their material, the authors are not quite sure how to finish the Discussion and have just listed some talking points. The purpose of this section needs to be reconsidered and the wording revised.
References:
Some formatting issues of the numbering are evident throughout.
Reviewer 3 Report
-A new section 2.1 should be introduced to present the experimental design if the study, in order to help potential readers comprehend the work.
-In the two figures, the various columns should be assigned different colours to facilitate understanding of the results.
-Tables or supplementary materials should be inserted, which will help readers to have a view about the real results.
-The discussion is verbose and should be reduced. Wording relevant to the New Zealand idiom should be replaced by proper english that everybody understands.
-Some relevant references are missing and should be included.
Round 2
Reviewer 3 Report
The authors have taken into account all the comments and have revised the manuscript correctly.
The manuscript is now acceptable for publication.